# Using Mendelian Randomisation to Prioritise Candidate Maternal Metabolic Traits Influencing Offspring Birthweight

**DOI:** 10.3390/metabo12060537

**Published:** 2022-06-10

**Authors:** Ciarrah-Jane Shannon Barry, Deborah A. Lawlor, Chin Yang Shapland, Eleanor Sanderson, Maria Carolina Borges

**Affiliations:** 1MRC Integrative Epidemiology Unit, University of Bristol, Bristol BS8 2BN, UK; d.a.lawlor@bristol.ac.uk (D.A.L.); chinyang.shapland@bristol.ac.uk (C.Y.S.); eleanor.sanderson@bristol.ac.uk (E.S.); m.c.borges@bristol.ac.uk (M.C.B.); 2Population Health Sciences, Bristol Medical School, University of Bristol, Bristol BS8 2PS, UK; 3NIHR Bristol Biomedical Research Centre, Bristol BS8 2BN, UK

**Keywords:** Mendelian randomisation, offspring, birthweight, genetics, maternal, metabolites, glucose, amino acids, lipids, nuclear magnetic resonance

## Abstract

Marked physiological changes in pregnancy are essential to support foetal growth; however, evidence on the role of specific maternal metabolic traits from human studies is limited. We integrated Mendelian randomisation (MR) and metabolomics data to probe the effect of 46 maternal metabolic traits on offspring birthweight (*N* = 210,267). We implemented univariable two-sample MR (UVMR) to identify candidate metabolic traits affecting offspring birthweight. We then applied two-sample multivariable MR (MVMR) to jointly estimate the potential direct causal effect for each candidate maternal metabolic trait. In the main analyses, UVMR indicated that higher maternal glucose was related to higher offspring birthweight (0.328 SD difference in mean birthweight per 1 SD difference in glucose (95% CI: 0.104, 0.414)), as were maternal glutamine (0.089 (95% CI: 0.033, 0.144)) and alanine (0.137 (95% CI: 0.036, 0.239)). In additional analyses, UVMR estimates were broadly consistent when selecting instruments from an independent data source, albeit imprecise for glutamine and alanine, and were attenuated for alanine when using other UVMR methods. MVMR results supported independent effects of these metabolites, with effect estimates consistent with those seen with the UVMR results. Among the remaining 43 metabolic traits, UVMR estimates indicated a null effect for most lipid-related traits and a high degree of uncertainty for other amino acids and ketone bodies. Our findings suggest that maternal gestational glucose and glutamine are causally related to offspring birthweight.

## 1. Introduction

Birthweight is frequently used as a proxy of foetal growth in population studies. Babies born with low or high birthweight are at increased risk of neonatal morbidity and mortality, and birthweight is inversely associated with risk of long-term adverse health outcomes, such as neurodevelopmental disorders and cardiometabolic diseases [1,2,3,4,5,6,7,8,9].

During pregnancy, the mother experiences marked physiological changes including substantial changes in maternal metabolism [10,11,12]. These changes are necessary to supply nutrients and oxygen to the foetus, enabling and sustaining healthy foetal growth and development. It is well known that maternal circulating glucose is transferred across the placenta to the foetus via facilitated diffusion and plays a vital role in supporting foetal growth [13,14,15,16,17]. Numerous other maternal metabolites, such as amino acids, fatty acids, ketone bodies, cholesterol, and glycerol, can cross the placenta and, therefore, potentially affect foetal growth directly [18]. In addition, some maternal metabolic traits may affect foetal growth indirectly (e.g., maternal triglycerides do not cross the placenta but are substrates for free fatty acids and ketone bodies) [17].

Previous observational epidemiological studies have suggested that maternal metabolic traits other than glucose (e.g., amino acids, triglycerides, cholesterol) are associated with birthweight [18,19,20,21,22,23,24,25,26,27]. For example, multiple studies have established a biologically plausible association between maternal lipids and foetal birthweight, independent of maternal glucose [18,19,20,26]. However, whether these findings reflect a causal effect of the maternal metabolite on birthweight is unclear since they are likely to suffer from residual confounding by multiple maternal lifestyle and health factors [28].

Mendelian randomisation (MR) is an instrumental variable analysis that uses genetic variants, usually single-nucleotide polymorphisms (SNPs), as instruments to infer causality in the presence of confounding [29,30,31]. MR has previously been used to explore the effects of a restricted set of maternal metabolic traits on offspring birthweight [32,33], which confirmed the causal effect of higher maternal glucose on higher mean offspring birthweight [32] but did not provide support for a role of maternal cholesterol in low- and high-density lipoprotein (HDLc and LDLc, respectively) or triglycerides on birthweight [32,33]. Recently a large MR study of 19 maternal circulating amino acids on foetal birthweight found evidence that glutamine and serine may have birthweight-increasing effects, alongside evidence that leucine and phenylalanine may have birthweight-lowering effects [34].

Recent technological advances in nuclear magnetic resonance (NMR) have enabled the metabolic profiling of large-scale population-based studies, including quantification of glucose, amino acids, ketone bodies, lipids, and many other metabolic traits [35,36]. This provides a unique opportunity to systematically explore the role of numerous maternal metabolic traits on birthweight. In this study, we integrated MR and NMR data to conduct a much broader interrogation of the effect of maternal metabolic traits on birthweight, using data from up to 210,267 individuals.

## 2. Results

### 2.1. Univariable MR

UVMR results obtained using SNPs selected from two independent genome-wide association studies (GWAS), i.e., the UK Biobank (UKBB) and Kettunen GWAS, are displayed in Figure 1; full results are given in Appendix A.

Using the SNPs selected from UKBB GWAS (K = 264 SNPs), UVMR of 46 metabolic traits identified three metabolic traits that fit our criteria and were included in MVMR (i.e., *p* < 0.05 in UVMR); higher maternal alanine (0.137 SD difference in birthweight per 1 SD difference in alanine (95% CI: 0.036, 0.239)), glutamine (0.089 (95% CI: 0.033, 0.144)), and glucose (0.328 (95% CI: 0.219, 0.436)) were associated with higher offspring birthweight. Estimates for ketone bodies were imprecise and had wide confidence intervals (CI). Most lipoprotein-related traits and fatty acids were precisely estimated around the null (Figure 1).

In the Kettunen GWAS (K = 40 SNPs), four metabolic traits were missing from these data and no SNPs were selected for another, leaving 41 out of the 46 metabolic traits. UVMR estimates using SNPs selected from Kettunen GWAS were largely consistent with the UKBB GWAS (Figure 1), although less precise, due to the lower sample size; hence, fewer SNPs were selected as instruments for most metabolites (Appendix A). Here, four metabolic traits met our criteria to be included in the MVMR analyses: higher maternal glucose (0.259 SD difference in birthweight per 1 SD difference in glucose (95% CI: 0.104, 0.414)), isoleucine (−0.127 (95% CI: −0.226, −0.029)), pyruvate (−0.135 (95% CI: −0.239, −0.031)), and 3-hydroxbutyrate (0.273 (95% CI: 0.144, 0.403)). We note that the birthweight-increasing effect of alanine observed in the UKBB GWAS was substantially attenuated when using Kettunen GWAS-selected SNPs. Additionally, there was a notable difference in the effect estimate for 3-hydroxybutyrate using SNPs selected from Kettunen compared to UKBB GWAS.

The mean F-statistics were calculated for each metabolic trait included in the UVMR analyses, as displayed in Appendix A. Mean F-statistics for each univariable model ranged from 54.863 (creatinine) to 296.941 (glycine) when using UKBB GWAS-selected SNPs and from 31.871 (lactate) to 100.951 (total lipids in very large HDL) for each univariable model when using Kettunen GWAS-selected SNPs.

When conducting leave-one-out analyses using UKBB-selected SNPs, we did not observe that any single SNP was driving UVMR estimates (Appendix A).

### 2.2. Multivariable MR

We selected putative causal metabolic traits to be included in MVMR: alanine, glucose, glutamine, isoleucine, pyruvate, and 3-hydroxybutyrate. MVMR results including all six maternal traits in a singular model are shown on Figure 2 and Appendix A.

First, we consider the results using UKBB-selected SNPs. Of the six selected candidate maternal metabolic traits in UVMR, MVMR analyses provided evidence of a direct causal effect on offspring birthweight for glucose (0.291 SD difference in birthweight per 1 SD difference in glucose (95% CI: 0.193, 0.389)), glutamine (0.074 (95% CI: 0.019, 0.130)), and possibly alanine (0.083 (95% CI: −0.016, 0.181)), although estimates for the latter were partly attenuated and less precise in MVMR compared to UVMR. The magnitude of the effects demonstrated was typically slightly larger in UVMR estimates than MVMR estimates, except for pyruvate.

We then consider the results using Kettunen-selected SNPs. Of the six selected candidate maternal metabolic traits in UVMR, MVMR analyses provided evidence of a direct causal effect on offspring birthweight for glucose (0.377 SD difference in birthweight per 1 SD difference in glucose (95% CI: 0.093, 0.661)). The estimated magnitude of effect was slightly attenuated relative to its corresponding UVMR estimate. The precision of estimates for higher maternal alanine (0.115 (95% CI: −0.231, 0.462)) and higher maternal glutamine (0.076 (95% CI: −0.106, 0.257)) was reduced in this dataset.

The conditional F-statistics were calculated for each of the MVMR selected models to test for the presence of weak instrument bias, Table 1.

The conditional F-statistics were calculated for the MVMR-selected models for both instrument selection approaches (Table 1). For the UKBB GWAS-selected SNPs, conditional F-statistics ranged from 11.406 (isoleucine) to 48.678 (glutamine), while, for Kettunen GWAS-selected SNPs, conditional F-statistics ranged from 6.120 (pyruvate) to 32.213 (glucose).

There was strong evidence of between-SNP heterogeneity in both MVMR models, $p=1.370\times 10^{-23}$ and $p=2.415\times 10^{-11}$ for the Q-statistic for the UKBB and Kettunen SNP sets, respectively.

### 2.3. Other Mendelian Randomisation Methods

For the six selected metabolites (i.e., alanine, glucose, glutamine, isoleucine, pyruvate, and 3-hydroxybutyrate), we compared our main UVMR estimates using IVW to from other UVMR methods (i.e., MR-Egger, weighted median, and weighted mode) that rely on different assumptions about the genetic instruments (Table 2 and Appendix A). We performed these additional analyses using UKBB-selected SNPs only (nine to 47 SNPs were selected for each metabolite) as the number of SNPs selected using Kettunen GWAS was often insufficient for such analyses (one to six SNPs selected for each metabolite). A similar pattern of results was found across the UVMR methods for glucose, with estimates found to be in the same direction as and of similar magnitude to those found using IVW. Estimates for the effect of glutamine and pyruvate were also found to be in the same direction (Table 2 and Appendix A), although estimates from the weighted median and mode approaches were attenuated for glutamine. For other metabolites (i.e., alanine, 3-hydroxybutyrate, and isoleucine) our main UVMR estimates were inconsistent with estimates of one or more of the alternative UVMR methods, although, in several instances, estimates were imprecise and their 95% confidence intervals were wide and overlapped (Table 2 and Appendix A).

## 3. Discussion

Our findings support a causal effect of some maternal metabolites on offspring birthweight. Taken together, our results confirm a causal effect of higher maternal glucose on offspring birthweight and suggest potential causal effects of higher glutamine. UVMR and MVMR results were supportive of an effect of higher maternal glutamine on higher offspring birthweight when using SNPs selected from UKBB GWAS, although results were less certain when using SNPs selected from Kettunen GWAS due to high imprecision resulting from the small number of SNPs being selected from the latter GWAS. This is in line with findings from a recent MR study probing the effect of 19 amino acids using large-scale data from a cross-platform metabolomics GWAS and from the same birthweight GWAS included in our study [34]. Glutamine is considered a conditionally essential amino acid in situations of physiological stress, such as pregnancy, when the (foetal) demand may exceed (maternal) synthesis [37]. The provision of amino acids from the maternal circulation to the foetus depends on placental amino-acid transporters, which are downregulated in pregnancies, resulting in foetal growth restriction [14,37,38,39]. Further glutamine is an important source of nitrogen and carbon to the foetus for protein synthesis and energy metabolism, as well as a precursor for the synthesis of other molecules needed to support foetal growth, such as nucleotides and glucosamines [40]. Initial UVMR analyses also suggested higher maternal alanine contributed to higher offspring birthweight; however, this effect was attenuated when using other UVMR methods in sensitivity analyses or when using SNPs selected from the Kettunen GWAS. UVMR analysis also identified isoleucine, pyruvate, and 3-hydroxbutyrate as putative causal metabolites; however, estimates were attenuated in MVMR or when using SNPs selected from UKBB.

In addition to prioritising putative metabolites influencing foetal growth, our study indicates that maternal lipid traits did not seem to contribute substantially to variations in offspring birthweight in this well-nourished population. This is supported by UVMR results, which estimated a null effect for most of lipid- and lipoprotein-related traits on birthweight. These findings are in line with previous MR studies of conventional clinical chemistry lipid traits (total HDL and low-density lipoprotein (LDL) cholesterol, and triglycerides [32,33]), but in disagreement with previously hypothesised effects of maternal triglycerides on offspring birthweight based on human observational studies and laboratory animal studies [19,20,26]. Furthermore, our study was unable to confirm the effects of maternal leucine and phenylalanine on decreasing birthweight and of serine on increasing birthweight as reported by a recent MR study [34], given that these were not picked up as causal in the UVMR metabolic trait selection procedure (i.e., leucine and phenylalanine) or were unavailable in our dataset (i.e., serine).

To our knowledge, this is the first study to integrate MR and high-throughput metabolomics data to (de)prioritise putative maternal metabolic factors influencing foetal growth. These findings could contribute to better understanding of key maternal molecular mechanisms influencing foetal growth.

One key challenge is determining which risk factors to include in the model. Including all available causal risk factors without careful consideration can be problematic due to the potential for bias due to the inclusion of highly similarly genetically predicted exposures or low power to identify the relevant risk factors when many risk factors which do not have a causal effect are included in the model [41]. We addressed such a challenge using a two-step procedure. First, we reduced the redundancy in the data by excluding highly related molecular traits and traits that were almost perfectly genetically correlated; thus, the genetically predicted values of the traits are indistinguishable. Second, we selected candidate metabolic traits for MVMR according to the results from UVMR, which is important to avoid diluting the predictive power of the underrepresented metabolite instruments through the inclusion of many strongly instrumented metabolic traits that were found to have no univariable relationship with the outcome. Although it is possible that bias in the UVMR due to pleiotropy could act in the opposite direction to a true causal effect to mask the effect of the metabolite on birthweight, such pleiotropy would have to be similar in magnitude to the causal effect. As our main concern in this analysis is pleiotropy acting through similar metabolic traits to the exposure, we consider it to be unlikely that such pleiotropy would act to mask a causal effect. MVMR analyses are generally of lower power than UVMR analyses; therefore, where no effect is observed in the UVMR due to low power, we would also not expect to be able to observe an effect for that metabolite in any MVMR analysis [41].

One key limitation of our study is the possibility that horizontal pleiotropy from traits not included in MVMR might have biased our results. The MVMR-IVW method will only provide a consistent estimate if all variants are valid instruments and not affected by unbalanced pleiotropy [41,42]. Given the interrelatedness of metabolites and their shared metabolic pathways, residual pleiotropy could plausibly bias our findings. Additionally, the calculated Q-statistics demonstrated strong heterogeneity within both MVMR models and inconsistencies across different UVMR methods, indicative that pleiotropy may potentially bias the causal effect estimates.

Another limitation of our study is the sample overlap between individuals in the risk factor and outcome datasets in the main analysis. Here, in the main analyses, UKBB participants were included in the SNP–risk factor and SNP–outcome estimates, with approximately ~30% of participants in the birthweight GWAS present in the UKBB metabolic traits GWAS. Selecting instruments in the same sample as the MR analysis may introduce a winner’s curse and overprediction; thus, we performed external validation and selected instruments from an independent dataset to assess the extent of the bias as a sensitivity analysis [43,44]. SNPs were identified for each available metabolic trait within the independent GWAS; then, summary data was extracted from the UKBB GWAS to minimise loss of power and maintain consistency of metabolic traits in the analysis. Our results consistently demonstrated statistical evidence that maternal glucose has a causal relationship with offspring birthweight using both UKBB and Kettunen GWAS-selected SNPs, which is reassuring given the well-established role of maternal glucose on offspring birthweight.

Furthermore, findings from this study should be interpreted bearing in mind that our sample mostly included individuals from European ancestry from high-income settings, which might limit the generalisability of the findings, and that the NMR platform used in this study is highly enriched for lipoprotein-related traits, whereas many other metabolic pathways are underrepresented. Thus, future research should focus on including more diverse populations and use higher-resolution metabolomics data, allowing for a broader interrogation of maternal metabolic traits. These will become feasible as the availability of large-scale GWASs in individuals of non-European settings and the use of mass spectrometry metabolomics increase. In addition, our findings are exploratory, and further MR studies investigating the mechanisms linking genetic variants to specific candidate metabolic traits (e.g., alanine and glutamine) are warranted to assess in depth the credibility of these findings and the plausibility of bias due to horizontal pleiotropy.

## 4. Materials and Methods

### 4.1. Overview

Figure 3 provides an overview of the study design. In brief, we selected genetic instruments for up to 46 NMR metabolic traits from two independent genome-wide association studies (GWAS), referred to as UK Biobank and Kettunen GWAS throughout, including up to 115,078 and 24,925 European ancestry individuals, respectively. We conducted two-sample univariable MR (UVMR) analysis to estimate the total causal effect of each NMR metabolic trait on offspring birthweight using the two sets of GWAS-selected genetic instruments. Genetic association data for metabolic traits and birthweight were extracted from UK Biobank (UKBB) and the 2019 Early Growth Genetics consortium (EGG) (N ≤ 210,267).

Importantly, genetic association data for offspring birthweight reflected direct maternal genetic effects (i.e., the effect of maternal genotype on offspring birthweight after accounting for potential bias from offspring genetic effects due to correlations between maternal and offspring genotypes) [45,46].

Many genetic variants regulate processes that affect multiple metabolites; thus, results from UVMR are likely biased by horizontal pleiotropy. Therefore, we used UVMR to identify candidate metabolic traits (*p* < 0.05), which were then taken forward into multivariable MR (MVMR) analyses. MVMR is an extension of UVMR that is suitable to jointly estimate the direct causal effect when considering risk factors that may be correlated, such as metabolic traits. In MVMR, multiple genetic variants that are potentially associated with several measured risk factors of interest are used to simultaneously estimate the direct causal effect of each risk factor on the outcome [47]. This accounts for pleiotropic effects via other risk factors included in the model which may bias UVMR estimates [48]. The effect estimated by MVMR is the direct effect not mediated by any other variable in the model.

### 4.2. Data Sources

Data were extracted from publicly available datasets of summary associations from GWAS consortia (Appendix A).

#### 4.2.1. Genetic Association Data for Metabolic Traits

We used genetic association data from two publicly available GWASs of metabolic traits available on the MR-Base data catalogue [49], generated as previously described. In both GWASs, metabolic traits were measured using targeted high-throughput NMR metabolomics (Nightingale Health Ltd., Helsinki, Finland), which provides simultaneous quantification of 249 metabolic traits (i.e., 165 metabolic traits and 84 derived ratios), encompassing routine lipids, lipoprotein subclass profiling (including lipid composition within 14 subclasses), fatty-acid composition, and various low-molecular-weight metabolites such as amino acids, ketone bodies, and glycolysis metabolites. Technical details and epidemiological applications were previously reviewed [36,50], and a full list of 249 metabolic traits is provided in Appendix A. Metabolic traits were standardised and normalised prior to analyses using rank-based inverse normal transformation (INT).

For the first GWAS, genetic association data were generated for up to 115,078 UK Biobank participants of European ancestry (54% females; age (years): mean = 56, SD: 8) for which metabolic traits were available using linear mixed model (LMM) association method as implemented in BOLT-LMM (v2.3) adjusting for genotype array, fasting time, and sex (hereafter ‘UKBB GWAS’), as previously described [51,52,53,54]. For the second GWAS, genetic association data were generated for up to 24,925 individuals of European ancestry from the Kettunen et al. meta-analysis including 10 studies adjusting for age, sex, time from last meal, and, where applicable, 10 first principal components [55] (hereafter ‘Kettunen GWAS’). The proportion of females in these studies ranged between 37% and 64%, with a mean age range of 31.2–61.3 years. The first dataset is better powered due to its larger sample size which enables the selection of a greater number of genetic instruments. However, there is sample overlap; the GWAS used for instrument selection includes UKBB participants in the genetic association data for both metabolic traits and offspring birthweight. Analyses using the Kettunen GWAS are likely to be underpowered due to its smaller sample size but are less likely to suffer from bias due to sample overlap as it does not include UKBB participants.

#### 4.2.2. Genetic Association Data for Offspring Birthweight

Genetic association data for offspring birthweight were extracted from a recent GWAS meta-analysis including 210,267 European participants, combining results of 41 studies from the EGG consortium and UKBB, with mean age at delivery ranging from 24.5 to 31.5 years (see Appendix B for further details) [56]. In this study, we used data from maternal genetic effects on offspring birthweight that was adjusted for offspring genotype, as previously described [56,57], in order to avoid potential biases due to the correlation between foetal and maternal genotypes [56,57].

### 4.3. Primary Exclusion Criteria for Metabolic Traits to Go into UVMR Analyses

The NMR platform used in this study includes many lipids and lipoproteins that are known to be closely (i) numerically and/or (ii) biologically related. As an example of (i), concentrations of cholesterol in different lipoprotein particles sum to the concentration of total cholesterol; therefore, total cholesterol is correlated with cholesterol in specific lipoproteins. As an example of (ii), the three glycolysis metabolites (glucose, lactate, and pyruvate) are metabolised via the same biological pathway (i.e., glycolysis), and, as a result, are correlated. This can result in variance inflation and biased estimates of effects in both observational and MVMR analyses if an attempt were made to mutually adjust for all or even a subset of them at the same time, unless one has extremely large sample sizes [58].

Therefore, we applied three exclusion criteria to the full list of 249 NMR metabolic traits to reduce redundancy in our UVMR analyses and avoid multicollinearity in our MVMR models (Appendix A).

First, we excluded those with multiple measures that reflected the same metabolic entity (e.g., same trait expressed as a proportion or concentration) or where multiple measures reflected a composite measure of highly related traits (e.g., ‘total fatty acids’ is a composite/combined measure of saturated, monounsaturated, and polyunsaturated fatty acids). As a result, we excluded 84 derived ratios and three composite fatty-acid measures (e.g., total fatty acids, polyunsaturated fatty acids, and degree of unsaturation).

Second, among lipoprotein-related traits, we selected measures related to circulating lipid composition (i.e., total triglycerides, phospholipids, esterified cholesterol, free cholesterol, phosphatidylcholines, and sphingomyelins), apolipoproteins A1 and B, and total lipids in 14 lipoprotein subclasses (i.e., extremely large, very large, large, medium, small, and very small very low-density lipoprotein (VLDL), IDL, large, medium, and small LDL, very large, large, medium, and small HDL). It has been demonstrated previously that we cannot conditionally predict different lipid measures within these subclasses in the same MVMR model due to the very high correlation between these traits [41]. This makes conducting UVMR on different elements within a subclass of lipoprotein redundant as they are not genetically separable and each UVMR would give equivalent estimated effects. As a consequence, we a priori chose to focus on total lipids in each subclass. This selection resulted in the exclusion of 114 measures, mostly representing lipid composition and particle concentration within lipoproteins/lipoprotein subclasses.

Third, we checked the genetic correlation across the remaining measures (Appendix A). If two traits are approximately genetically identical, they cannot be distinguished in UVMR analysis or analysed as two distinct traits in a MVMR analysis, due to multicollinearity. Therefore, if a pair of metabolic traits was found to be highly genetically correlated ($r^{2}$ > 0.985), one element of the pair was removed. This resulted in the exclusion of two additional metabolic traits (“total lipids in small LDL” and “total lipids in very large VLDL”).

After all exclusions, detailed in Appendix A, 46 metabolic traits remained available for analyses (Appendix A).

### 4.4. Statistical Analyses

We estimated the effects of maternal metabolic traits on offspring birthweight using UVMR and MVMR. Genetic association data for metabolic traits and offspring birthweight were harmonised to reflect the same effect allele using allele frequency information to infer the strand for palindromic SNPs [59]. Palindromic SNPs with MAF > 42% were considered ambiguous and removed from analyses. All statistical analyses were performed using R (version 4.0.3, the R Core team, Boston, MA, USA) using the ‘two-sample MR’ R package (version 4.0.2, Bristol, UK) [49] and the ‘MVMR’ R package (version 1, Bristol, UK) [41].

### 4.5. Univariable MR

We selected SNPs strongly and independently associated with each of our selected metabolic traits using a threshold of $p<5\times 10^{-8}$ and $r^{2}<0.01$ (using 1000 Genomes EUR population for estimating pairwise linkage disequilibrium (LD)) using data from the UKBB and Kettunen GWASs. Then, we used UVMR to combine genetic association data for metabolic traits (from UKBB) and birthweight (from EGG + UKBB) using SNPs selected from UKBB and Kettunen GWASs. We applied an automated approach for the method of analysis, contingent on the number of associated SNPs for each metabolite; 1 SNP—Wald ratio estimate, >1 SNPs—IVW estimate.

We present UVMR results according to subclasses of metabolic traits (amino acids, apolipoproteins, cholesterol esters, fluid balance, inflammation, ketone bodies, lipoprotein subclasses, other lipids, phospholipids, and triglycerides).

To test instrument strength, we calculated the mean F-statistics across SNPs for each metabolic trait included in the UVMR analyses [41,47].

We also performed leave-one-out analyses, sequentially omitting the indicated SNP and rerunning the analysis. This may indicate whether the estimated association is being substantially driven by a singular SNP.

### 4.6. Multivariable MR

All metabolic traits that demonstrated statistical evidence of an effect in the UVMR (p<0.05) were included in the MVMR estimation of the direct effect on birthweight (using either of UKBB and Kettunen GWAS-selected SNPs).

We then applied MVMR to estimate the direct effect on birthweight of all candidate metabolic traits selected on the basis of UVMR results [47]. All SNPs associated with at least one of the candidate metabolic traits were selected for the MVMR analysis. An additional round of clumping was applied to ensure no genetic variants for one metabolic trait were in LD with genetic variants for any other metabolic trait.

To test instrument strength, the conditional F-statistic was calculated, which is the multivariable equivalent to a standard F-statistic; thus, a conditional F-statistic greater than 10 in MVMR implies that the model is unlikely to suffer from substantial weak instrument bias [47]. A phenotypic correlation matrix derived from NMR UKBB data was used to approximate the pairwise covariances for the SNP–metabolic trait associations in the calculation of conditional F-statistics.

Outlying instruments in MVMR analysis were quantified by the Q-statistic, calculated from the *p*-value-adjusted χ^2^ distribution. Degrees of freedom were determined by the number of instruments available for MVMR analyses, minus the number of metabolic traits, minus one [41].

### 4.7. Other Mendelian Randomisation Methods

We applied alternative UVMR estimation methods that are reliant upon different assumptions to explore the sensitivity of our main UVMR estimates to the IVW method’s assumptions. MR-Egger, weighted median, and weighted mode were used to explore the presence of horizontal pleiotropy and the robustness of our results to the presence of invalid instruments [60,61,62]. If the magnitude and direction of effects are inconsistent with the estimates obtained via IVW, this is indicative of bias due to invalid instruments. MR-Egger can estimate causal effects in the presence of invalid instruments provided that the magnitude of the pleiotropic effect is unrelated to the strength of the association between the instruments and the exposure (known as the INSIDE assumption) [61]. The weighted median estimate provides a consistent estimate of the causal effect when at least 50% of the weight in the analyses comes from valid instruments [60]. Similarly, but less restrictive than weighted median, the weighted mode estimate requires the largest subset of instruments to be valid in which they identify a homogeneous causal effect estimate [62].

## 5. Conclusions

In summary, in this study, we demonstrated the utility of integrating MR and metabolomics data to (de)prioritise candidate maternal molecular traits influencing foetal growth. Our findings confirm previous studies on the strong relation between maternal glucose and offspring birthweight, highlight glutamine as a potential causal metabolite, and do not support a relationship between several maternal lipids and offspring birthweight. Identification of specific metabolic traits represents a unique opportunity to understand the relation between the intrauterine environment and offspring growth.

## Figures and Tables

**Figure 1 metabolites-12-00537-f001:**
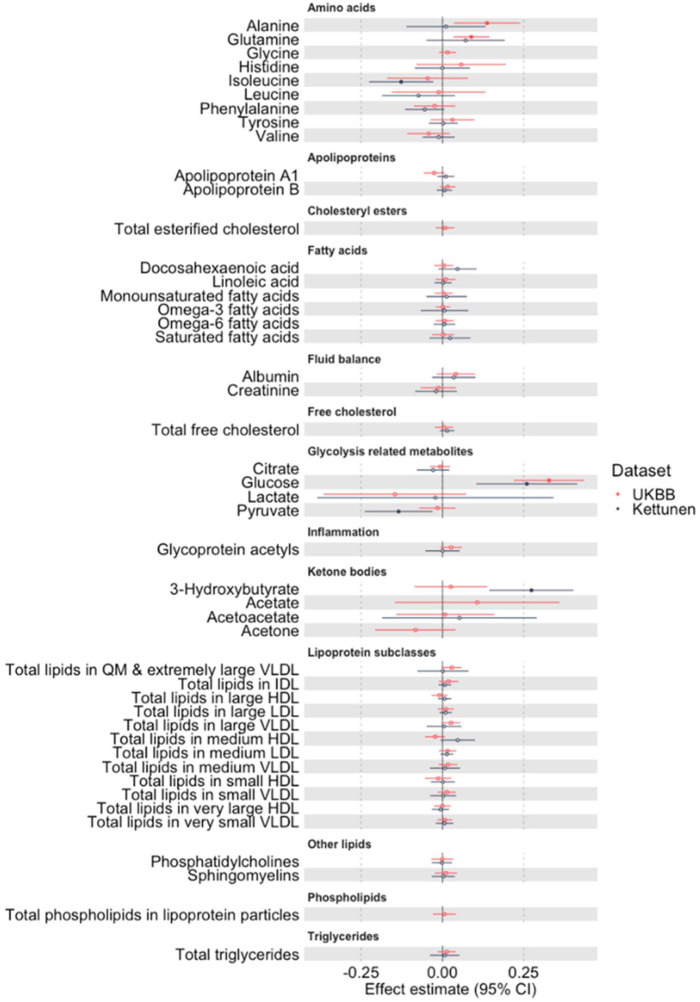
UVMR effect estimates, with 95% confidence interval (CI), for the relation of maternal metabolic traits with offspring birthweight using SNPs selected from UKBB and Kettunen GWAS. Effect estimates (with 95% CI) are expressed as standard deviation (SD) units of offspring birthweight per SD unit change in metabolic trait using the Wald ratio (no. SNPs = 1) or IVW (no. SNPs > 1). The number of SNPs selected from UKBB and Kettunen GWAS for each metabolic trait is available in Appendix A. Solid circles identify the metabolic traits that were taken forward to MVMR on the basis of having some evidence for statistical association with birthweight in UVMR (*p* < 0.05). CI: confidence interval, GWAS: genome-wide association study, MVMR: multivariable Mendelian randomisation, SNP: single-nucleotide polymorphism, UKBB: UK Biobank, UVMR: univariable Mendelian randomisation.

**Figure 2 metabolites-12-00537-f002:**
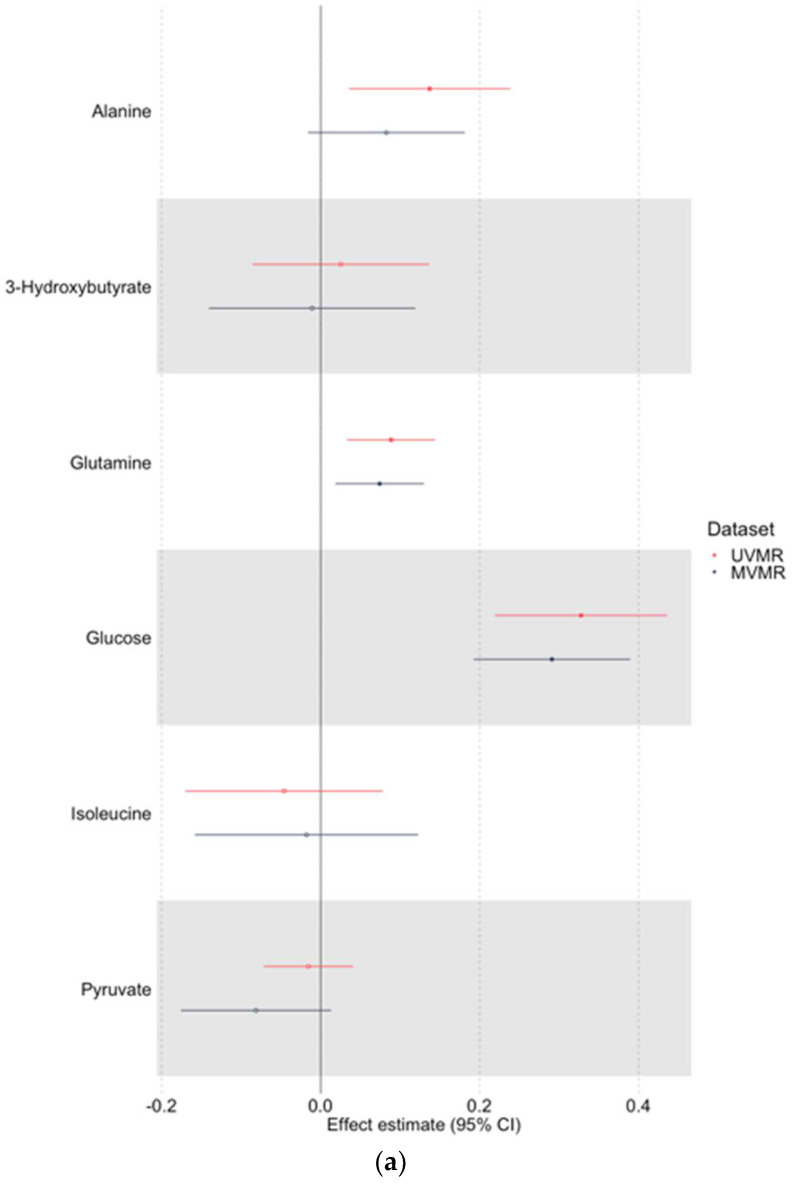
UVMR and MVMR effect estimates, with 95% confidence interval (CI), for six maternal metabolic traits (alanine, glucose, glutamine, isoleucine, pyruvate, and 3-hydroxybutyrate) on birthweight in SNPs selected from UKBB (**a**) and Kettunen (**b**) GWAS. UVMR estimates were calculated using the Wald estimate (no. SNPs = 1) or IVW (no. SNPs > 1). All MVMR estimates were calculated using IVW. CI: confidence interval, GWAS: genome-wide association study, IVW: inverse variance weighting, MVMR: multivariable Mendelian randomisation, SNP: single-nucleotide polymorphism, UKBB: UK Biobank, UVMR: univariable Mendelian randomisation.

**Figure 3 metabolites-12-00537-f003:**
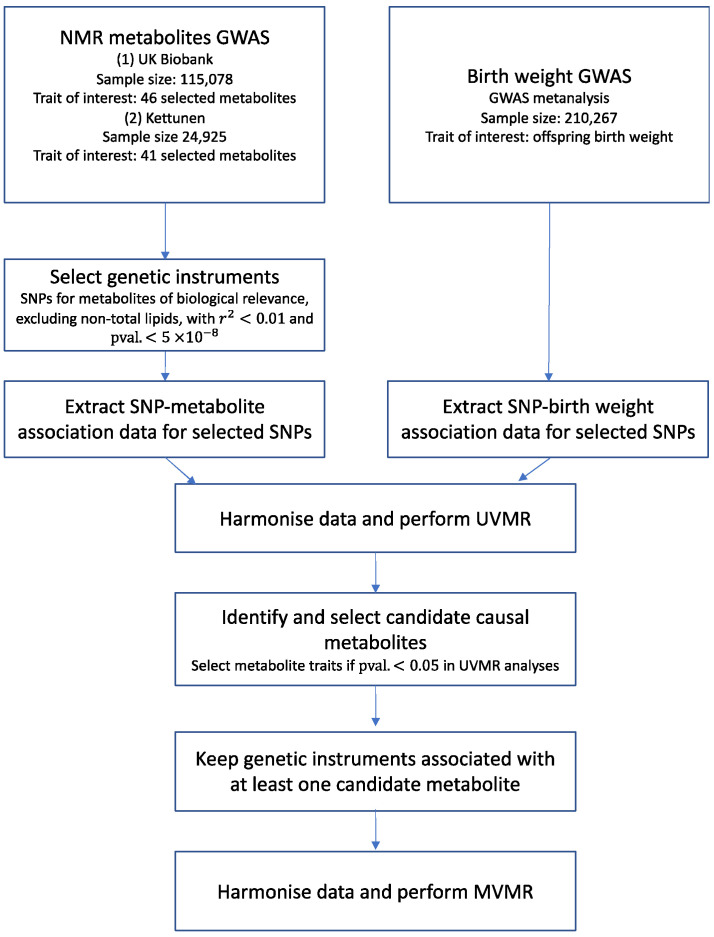
A summary of methods used to select and estimate effects of metabolic traits on offspring birthweight in the Mendelian randomisation analysis. GWAS: genome-wide association study, MVMR: multivariable Mendelian randomisation, NMR: nuclear magnetic resonance, SNP: single-nucleotide polymorphism, UKBB: UK Biobank, UVMR: univariable Mendelian randomisation, *r^2^*: genetic correlation between SNPs.

**Table 1 metabolites-12-00537-t001:** The conditional F-statistics for the metabolic traits included in MVMR main analysis.

Metabolite	Conditional F-Statistics (UKBB-Selected SNPs, K = 84)	Conditional F-Statistics (Kettunen-Selected SNPs, K = 12)
Alanine	20.292	21.723
3-Hydroxybutyrate	12.986	10.823
Glutamine	48.678	12.603
Glucose	20.179	32.213
Isoleucine	11.406	7.011
Pyruvate	22.916	6.120

K is the number of selected SNPs (K). The conditional F-statistics for both analyses were calculated using summary genetic data for metabolic traits from UK Biobank. CI: confidence interval.

**Table 2 metabolites-12-00537-t002:** UVMR estimates obtained from a range of estimation methods as a sensitivity test to explore the presence of horizontal pleiotropy.

	MR-Egger	Weighted Median	Weighted Mode		MR-Egger	Weighted Median
Metabolite	Effect Estimate (95% CI)	*p*	Effect Estimate (95% CI)	*p*	Effect Estimate (95% CI)	*p*
Alanine	0.003 (−0.255, 0.261)	0.982	0.009 (−0.073, 0.091)	0.837	−0.051 (−0.139, 0.037)	0.267
3-Hydroxybutyrate	0.207 (−0.075, 0.489)	0.171	−0.058 (−0.164, 0.049)	0.287	−0.147 (−0.371, 0.077)	0.217
Glutamine	0.079 (−0.007, 0.165)	0.079	0.037 (−0.004, 0.079)	0.078	0.026 (−0.015, 0.067)	0.220
Glucose	0.279 (0.034, 0.524)	0.037	0.379 (0.297, 0.461)	<0.001	0.374 (0.289, 0.458)	<0.001
Isoleucine	−0.179 (−0.545, 0.187)	0.371	−0.023 (−0.139, 0.093)	0.697	0.042 (−0.137, 0.221)	0.656
Pyruvate	−0.061 (−0.171, 0.049)	0.294	−0.048 (−0.115, 0.02)	0.167	−0.063 (−0.138, 0.012)	0.118

## Data Availability

Data used in this study are publicly available. Genetic association data on birthweight were contributed by the EGG Consortium using the UK Biobank Resource and were downloaded from www.egg-consortium.org, accessed date 21 January 2021. Genetic association data on NMR metabolic traits from UK Biobank participants were generated as previously described under UK Biobank Project 30418 [52,53]. Genetic association data on NMR metabolic traits from Kettunen et al. were extracted from the IEU Open GWAS platform [63].

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
