# Peer review of "Using Mendelian Randomisation to Prioritise Candidate Maternal Metabolic Traits Influencing Offspring Birthweight"

_metabolites, 2022, doi:10.3390/metabo12060537_

Round 1
Reviewer 1 Report
Authors investigated the impact of maternal metabolites on the offspring birth weight through mendelian randomization. The paper is well written and main results are clearly outlined in main figures and tables. The authors properly accounted for the main sources of bias in their analysis : the correlation between maternal and offspring genotypes and horizontal pleiotropy. While the association between maternal blood glucose and birth weight is well known, the potential associations between amino acid are novels (to the extent of reviewer knowledge). The paper also provides clarification on the lack of support for maternal lipids. The report seems suited for the journal. Nonetheless, a few concerns should be addressed prior to publication.
The reviewer states the following concerns :
Major :
- No sufficient justifications were provided for the choice of the MVMR methods. The application of another method such as MR-PRESSO (Verbanck et al) or MR-Egger (Bowden et al) with a comparison of the outcomes would have been welcomed. At least a theoretical argument should be provided on why the authors preferred the method they developed over well established ones in this field.
- Systematically report p-value along with statistics in tables.
- Clarify if multiple testing corrections were taken into account in reported MVMR associations.
Minor :
- Merge Table S1 and S2 to allow reader to easily compare estimates in UKBB and Kettunen
Author Response
Please see the attachment, thank you.

Reviewer 2 Report
This is an interesting paper, I would like the authors to carefully proof check the whole article as there are many typos or errors as below.
Minor comments:
In section “2.1. Univariable MR”, I think you are talking about Figure 1, but you mentioned Figure 2 in lines 90 and 95.
Any explanation for “Estimates for ketone bodies were imprecise and had wide confidence intervals (CI)”? Due to the small sample size? But this is for big UKBB not small Kettunen, right?
The p-value was written in the wrong way in line 150.
You have “4.3.1”., but no “4.3.2”.
In Tables S1 and S2, the authors don’t need to show SE as the 95% CI has already been shown. Please present a P-value instead of SE.
In Table S3, I also would like to know how many overlapped SNPs for each trait between UKBB and Kettunen. Add a new column in the table?
Author Response
Please see the attachment, thank you.

Reviewer 3 Report
Barry et al. aims to question whether where is any correlations between specific maternal metabolic traits and fetal growth in addition to circulating glucose levels. The authors clearly summarize previous observations and integrate a sensitive statistical methods including Mendelian randomization (MR) and metabolomics data to probe the effect on metabolic traits on fetus growth. The manuscript is written precisely and data presented is significantly depicted. Methods described clearly. Therefore i suggest to accept the manuscript at the present form.
Author Response
Please see the attachment, thank you.

Round 2
Reviewer 1 Report
I thank the author for addressing all my comments and reporting UVMR estimates from additional methods. While the report is thorough and detailed, the evidence for the main novelty of the article (association between maternal amino acids and offspring birth weight) is quite limited in light of these new analysis. I suggest that the abstract should be clarified on that point. While few significant associations were detected, I think the report is worth publishing as it is scientifically sound and the lack of causal association between numerous maternal metabolites and offspring weight is in itself a results.
